# Managing Microbiota Activity of *Apis mellifera* with Probiotic (Bactocell^®^) and Antimicrobial (Fumidil B^®^) Treatments: Effects on Spring Colony Strength

**DOI:** 10.3390/microorganisms12061154

**Published:** 2024-06-06

**Authors:** Joy Gaubert, Pierre-Luc Mercier, Georges Martin, Pierre Giovenazzo, Nicolas Derome

**Affiliations:** 1Derome Laboratory, Institut de Biologie Intégrative et des Systèmes, Département de Biologie, Université Laval, Québec, QC G1V 0A6, Canada; pierre-luc.mercier.1@ulaval.ca (P.-L.M.); nicolas.derome@bio.ulaval.ca (N.D.); 2Giovenazzo, Laboratory, Département de Biologie, Université Laval, Québec, QC G1V 0A6, Canada; pierre.giovenazzo@bio.ulaval.ca; 3Centre de Recherche en Sciences Animales de Deschambault, Deschambault, QC G0A 1S0, Canada; georges.martin@crsad.qc.ca

**Keywords:** honeybee, microbiota, probiotic, Bactocell^®^, Fumidil B^®^, colony wintering

## Abstract

Against a backdrop of declining bee colony health, this study aims to gain a better understanding of the impact of an antimicrobial (Fumidil B^®^, Can-Vet Animal Health Supplies Ltd., Guelph, ON, Canada) and a probiotic (Bactocell^®^, Lallemand Inc., Montreal, QC, Canada) on bees’ microbiota and the health of their colonies after wintering. Therefore, colonies were orally exposed to these products and their combination before wintering in an environmental room. The results show that the probiotic significantly improved the strength of the colonies in spring by increasing the total number of bees and the number of capped brood cells. This improvement translated into a more resilient structure of the gut microbiota, highlighted by a more connected network of interactions between bacteria. Contrastingly, the antimicrobial treatment led to a breakdown in this network and a significant increase in negative interactions, both being hallmarks of microbiota dysbiosis. Although this treatment did not translate into a measurable colony strength reduction, it may impact the health of individual bees. The combination of these products restored the microbiota close to control, but with mixed results for colony performance. More tests will be needed to validate these results, but the probiotic Bactocell^®^ could be administrated as a food supplement before wintering to improve colony recovery in spring.

## 1. Introduction

The honeybee (*Apis mellifera*) is one of the most domesticated insects [1]. It is a key species as a pollinator for the agri-food and pharmaceutical industries, thanks to the hive products [2,3]. Nowadays, honeybee colony health is a growing source of concern. Beekeepers are suffering increasing losses, often characterized by high winter mortality, especially in North America and Europe [4,5]. The cause of this declining health is probably a combination of several stressors affecting colonies simultaneously. These stressors may be biotic (e.g., *Varroa destructor* parasite and deformed wing virus) or abiotic (e.g., antifungals, pesticides), and disturb the functioning of individual bees or the colony as a whole [6]. They can interact with each other, inflicting additive, synergistic, or antagonistic effects on bees and colonies [6,7,8]. It is therefore necessary not only to measure the effects of each stress factor in order to identify the main threats, but also to seek solutions capable of offsetting their complex synergistic effects. Indeed, honeybee behavior and defense systems are impacted by these stress factors [9,10] as are their gut microbial symbionts [11].

The microbiota can be defined as all the microorganisms living in symbiosis on the host’s tissues. Healthy bees host a stable community of microorganisms, carrying out beneficial functions for them, a state known as eubiosis [12]. In this case, taxonomic composition of the microbiota is essentially restricted and conserved within the host species, consisting of five main bacterial taxa that define the core microbiota (*Snodgrassella alvi*, *Gialliamella apicola*, *Bombilactobacillus* sp. (formerly named *Lactobacillus Firm-4*), *Lactobacillus* sp. (formerly named *Lactobacillus Firm-5*), and *Bifidobacterium asteroides*), often completed by four other taxa (*Bartonella apis*, *Apibacter adventoris*, *Frischella perrara*, and *Acetobacteraceae*) [13]. The gut bacterial microbiota plays key functions for its host such as synthesizing molecules (enzymes, vitamins, etc.), activating the immune system, and assimilating nutrients [14,15,16]. However, stress factors induce overall changes in the bacterial environment, therefore disrupting the equilibrium between beneficial, neutral (core members, commensals), and opportunistic and/or pathogenic strains in the microbiota. When linked to diseases, this phenomenon is called dysbiosis [17,18]. Gut dysbiosis can lead to alternate microbiota profiles associated with non-thriving colonies or diseases states [19]. Parasites or pesticides can induce such dysbiosis [20]. This phenomenon was also observed with veterinary treatments administrated by beekeepers. For example, micro-hives exposed to three of these treatments (oxytetracycline, sulfonamides, and tylosin) showed changes in microbiota taxa abundances and diversity, as well as an increase in the rate of tetracycline resistance genes among bacteria [21].

The aim of this study was to determine the potential effects of antimicrobial treatment and probiotic supplementation, administered individually and in combination, in autumn, on the health of honeybee colonies and their microbiota during wintering and early spring. Indeed, by improving bee health at the individual level, a balanced microbiota also improves colony performance, since healthy bees are more numerous (less mortality), have fewer hygienic tasks to perform (e.g., evacuating dead individuals), and can therefore allocate their full capacity to carrying out the main tasks of the colony, including pollination and honey production. Carried out on a colony scale, this study produced realistic results and observations that are transferable to industrial applications. In this context, the first product selected for this study was the Fumidil-B^®^ (whose active ingredient is the Fumagiline B), used by beekeepers to control the intestinal microsporidia *V. ceranae*. Although fumagillin degrades over time, its breakdown is highly dependent on environmental conditions [22]. Moreover, beekeepers must perform repeated and seasonal treatments against *Vairimorpha* spp. [23]. Therefore, multiple generations of bees could be exposed to low doses of fumagillin. Like tetracycline, used against European and American foulbroods [24], Fumidil B^®^ might create dysbiosis with consequences for bee health such as impaired metabolism and weakened defenses systems [14,15,16]. Therefore, this study could also highlight the effects of gut dysbiosis on the performance of functions provided by the microbiota. The Fumidil B^®^ active ingredient acts by binding and inhibiting the methionine aminopeptidase type 2 (METAP-2) [25]. This ubiquitous enzyme has a key function in tissue repair and protein degradation. Fumagiline B may have chronic effects on honeybees as it may act on METAP-2 in the same way as it acts against *Vairimorpha* spp., as the binding sites of fumagillin appear to be the same for *Vairimorpha* spp. and bees [26]. It has been shown that Fumagiline B induces alterations in the structural and metabolic proteins of the honeybee midgut at concentrations that do not suppress parasite multiplication [26]. It has also been shown to reduce honeybee lifespan [27,28].

Given the microbiota importance for host health, restoring honeybee microbiota after exposition to such stressors could improve their resilience, as well as increasing their global performance. It is therefore necessary to consider sustainable tools to prevent or mitigate the dysbiosis state in bee colonies. Probiotic supplementation could fulfill this mandate. A probiotic is defined as live microorganisms that, when given in the right quantities, provide a health benefit to the host [29]. The probiotic and second product selected for this study is Bactocell^®^ (Lallemand, Inc.). This probiotic consists of the bacterial strain *Pediococcus acidilacticis*. This bacterium is a member of lactic acid bacteria (LAB), able to adapt to the intestinal environments of several types of host. Experimental administration of this probiotic showed promising results by improving bee survival when exposed to *V. ceranae* spores in tests carried out with caged bees [30]. However, these results still need to be validated at the apiary scale, given that both social aspects and complex nutrition in honeybee colonies play an important role in treatment response [6,31].

In this study, honeybee colonies were exposed in autumn to the antimicrobial Fumidil B^®^ and the probiotic Bactocell^®^ to measure to what extent these products may have deleterious and/or beneficial effects on winter survival and spring recovery. At the same time, honeybee gut microbiota collected from these colonies at the end of treatment (before wintering) were analyzed to determine the treatment effect on honeybee microbiota integrity. Our working hypothesis was that both treatments and their combination would change the microbiota structure and correlate with differences regarding hive performance after wintering. Specifically, three hypotheses were tested: (1) Fumagilin will alter the microbiota composition and its resiliency (dysbiosis) and decrease colony performance; (2) Bactocell^®^ will maintain the microbiota composition and its resiliency (eubiosis) and enhance colony performance; and (3) combining Fumidil-B^®^ and Bactocell^®^ (group FB) will maintain or restore the microbiota composition and its resiliency (eubiosis) without having a negative impact on colony performance.

## 2. Materials and Methods

No animal health care permits were required for this study. This study took place at the Centre de Recherche en Sciences Animales de Deschambault (CRSAD, Deschambault, QC, Canada; 46 40030.000 N, 71 54052.300 O) between 7 October 2021 and 30 May 2022. On 15 July 2021, honeybee colonies were prepared with young sister queens to reduce the influence of genetic diversity across experimental groups in this experiment, then placed in two honey-producing apiaries located on farmland near the research facility. At the beginning of September, honey supers were removed, and colonies were reduced to one brood chamber. Fall feeding started in mid-September and all colonies were given 24 L of 2:1 sucrose solution using a top box feeder (Wooden Miller feeder # FE-1100 from Propolis-etc., Beloeil, QC, Canada). Colonies received a Thymovar^®^ anti-varroa treatment starting on 9th September, followed by an oxalic acid treatment on 2nd November (drip method: 35 g/L in a sucrose 1:1 solution, 5 mL between every frame of the hive body crowded with honeybees, to a maximum of 50 mL per colony). Prior to the experiment (October 7), the strength of the colonies was measured by weighing and by visually estimating the frame area covered with brood [32]. Then, experimental groups were formed to be of similar strength, in particular, average hive weight and average number of bees and brood per colony were equivalent between groups (Appendix A). Colonies were wintered indoors in an environmentally controlled room (4–5 °C, 50–60% RH) from 15 November to 20 April and then moved to two spring apiaries on 5 May before final brood evaluation on 30 May. These steps are summarized in Figure 1.

### 2.1. Experimental Design

A total of 48 colonies (12 colonies by group) from two apiaries were used for this project to enhance the statistical power. The colonies of each group were randomly allocated to minimize selection bias. After evaluation (brood and adult bee population, pollen and nectar resources), they were divided into four groups of similar colony structure and distributed equally and randomly between two apiaries. The treatments were randomly distributed to the groups (Table 1). The groups of 12 colonies were as follows: the first group (control) was the control and received a sucrose solution 1:1 (*w*/*w*) (called syrup for the rest of the article); the second group was treated with Bactocell^®^ (syrup containing 10^8^ CFU of Bactocell^®^); the third group was treated with Fumidil B^®^ (syrup containing 1.25 × 10^−3^ g of Fumidil B^®^); and the fourth group received both Fumidil B^®^ and Bactocell^®^ (FB, syrup containing 1.25 × 10^−3^ g of Fumidil B^®^ and 10^8^ CFU of Bactocell^®^).

### 2.2. Treatments

The control treatment solution contained a sugar syrup 1:1 (one volume of sugar for one volume of water). The probiotic treatment solution was a mixture of 1 L of 1:1 syrup and 10 g of Bactocell^®^ powder (Lallemand Inc., Montreal, QC, Canada) at 1 × 10^10^ UFC/g. The antibiotic treatment solution was a mixture of 1 L of 1:1 syrup for 1.25 g of Fumidil B^®^ (Fumagilin-B, Can-Vet Animal Health Supplies Ltd., Guelph, ON, Canada) and the combination of probiotic and antibiotic was made of 1 L of 1:1 sugar syrup and 10 g of Bactocell^®^ powder at 1 × 10^10^ UFC/g and 1.25 g of Fumidil B^®^.

### 2.3. Treatments Administration

The colonies were treated four times in the autumn, one week apart (on 7, 14, 21, 28 October) before the colonies were placed in the wintering room (on 15 November). Colonies of various groups received 50 mL of appropriate treatment solution, which was dripped with a syringe between hive frames and on bees (5 to 10 mL per frame of bees) to encourage mutual cleaning behavior and the dispersion of the product throughout the colony. Fumidil B^®^ was given only for the first application on 7 October, according to veterinary and product recommendations. For the three subsequent applications, the Fumidil B^®^ group received only syrup and the Fumidil B^®^ + Bactocell^®^ group received syrup with Bactocell^®^ only. After wintering, the same treatments were repeated on 3, 11, 18, and 25 May).

### 2.4. Sampling

Approximately 100 honeybees were sampled in each hive, once before the treatments (t0, 7 October) and once after the fall treatments (t21, 28 October). After securing the queen, bees were collected from a broodless frame in a 50 mL Eppendorf tube. Samples were immediately put on dry ice and stored at −86 °C until lab analysis (Thermofisher −86 °C FORMA 908, Waltham, MA, USA). Sampling was carried out twice, before exposure to the treatments (t0) and after 21 days (t21).

### 2.5. Measurement of Zootechnical Performance Traits

Winter survival: Colony survival was noted in April when the hives were moved from the overwintering chamber. A colony without brood or with fewer than two frames covered in bees was considered dead.

Brood population: The area occupied by immature worker honeybees (eggs + larva + sealed brood) in colonies was evaluated by measuring the width and length of the brood area on each side of every brood frame. The obtained rectangular area was multiplied by 0.8 to compensate for the elliptic form of the brood pattern [32]. These values were added to calculate the total brood area in each colony. A factor of 25 worker cells per 6.25 cm^2^ (i.e., a square inch) was used to calculate the number of immature workers from the area. Brood cells are visually distinguishable and are separated into two categories: some are open (unsealed brood cells), and others are covered with a porous wax lid (sealed brood cells). This calculation was performed first on 7 September 2021, to create equal experimental groups, then on 30 May 2022, to evaluate the colonies after wintering.

Weight: Hives were weighed on 7 October and 25 April, using a numeric platform scale (total capacity of 500 kg, minimum weight sensitivity of 0.1 kg). The winter weight loss is the difference between these two weights for each hive.

Honeybee population: The size of the bee cluster was measured in October and April, before and after the overwintering period. The number of frames covered by the bees was recorded, both from above and below, and the mean value was calculated for each colony.

Rate of bee loss during wintering: The number of bees was determined in October and April. To consider the variability of each colony in terms of strength, the percentage of loss for each colony was calculated.

*Varroa destructor* infestation levels: On 3 May 2022, *Varroa destructor* mite population was monitored by sampling nurses (on a frame of capped brood). For this, nurses were swept (bee broom) and placed in 120 mL plastic jars containing 80 mL ethyl alcohol using a funnel until the jar was full. In the laboratory, pots of bees were then placed on a rotating orbital agitator for 5 min at 130 rpm. Their content was then poured over a container topped with a mesh screen to separate the bees from the ethanol now containing the varroa mites. Fifty bees and the total number of bees in the sample were then weighed to determine the number of bees in each sample. The cycle was repeated until no more bees could be seen in the sample, or after the third cycle. The infestation rate was then calculated as the number of varroa mites per 100 bees. This variable was included in the statistical analyses, as it is known to influence colony performance [33].

These parameters were compared between groups using ANOVA or Kruskal–Wallis analysis (depending on their conditions of normality and homogeneity of variances).

### 2.6. Construction of 16S rRNA Gene Libraries

#### 2.6.1. DNA Extraction

The sampled bees were dissected under flame and with alcohol-sterilized instruments to prevent external contamination. After collecting the midgut, DNA was extracted using the salt method described by Aljabani and Martinez (1997) [34].

#### 2.6.2. First DNA Amplification

V3-V4 hypervariable regions were amplified in a PCR1 using primers at 10 mM (347-F: 5′-ACACTCTTTCCCTACACGACGCTCTTCCGATCT-GGAGGCAGCAGTRRGGAAT-3′ and 803-R: 5′-GTGACTGGAGTTCAGACGTGTGCTCTTCCGATCT-CTACCRGGGTATCTAATCC-3′) (Sigma-Aldrich Life Science, Oakville, ON, Canada). The master mix was made using the following instructions: for 100 µL of master mix: reaction buffer (Q5) (5×) 20 µL; dNTPs (10 mM) 2 µL; 347-F 5 µL; 803-R 5 µL; high GC enhancer (Q5) (5×) (NEB) 20 µL; Q5 high-fidelity DNA polymerase (NEB) 1 µL; H_2_O 41 µL; and DNA template 3 µL. The PCR program was as follows: denaturation at 98 °C for 2 min, amplification during 30 cycles of 10 s at 98 °C, 30 s at 60 °C, and 30 s at 72 °C and extension at 72 °C for 2 min. Final products were run on 2.0% agarose gels to detect fragments of the size of 450 bp and quantified using a Nanodrop 1000 spectrophotometer (Thermo Fisher, Ottawa, ON, Canada). Amplification results were then purified by adding 70% of their volume on AMPure beads (Beckman Coulter Genomics, Mississauga, ON, Canada), followed by five minutes’ incubation at ambient temperature. Once placed in a magnetic rack, another five minutes’ incubation at ambient temperature was performed and the liquid was removed. Still in the magnetic rack, 200 µL of 80% fresh ethanol was then added and removed after 30 s (two times). The beads were left to dry for four minutes and, after removing samples out of the magnetic rack, 22 µL of DNase free water was added. After two minutes of incubation, samples were placed again in the magnetic rack for a five-minute incubation. An amount of 20 µL of the liquid was then transferred to a new tube for each sample.

#### 2.6.3. Second Amplification

A PCR2 was performed to attach the barcode sequences needed for the sequencing. Illumina dual-Indexes came from the Plateforme d’Analyses Génomiques (IBIS, Université Laval). Each sample needed to have a unique barcode combination. For a volume of 50 µL of master mix, it contained the following: reaction buffer (Q5) (5×) 10 µL; dNTPs (10 mM) 1 µL; index-F 1 µL; index-R 1 µL; high GC enhancer (Q5) (5×) (NEB) 10 µL; Q5 high-fidelity DNA polymerase (NEB) 0.5 µL; H_2_O 24.5 µL; and DNA template 2 µL. The PCR program was as follows: denaturation at 98 °C for 2 min, amplification during 9 cycles of 10 s at 98 °C, 30 s at 60 °C, and 30 s at 72 °C and extension at 72 °C for 2 min. The PCR results were purified after the same protocol used after the first amplification and pooled in a 1.5 mL Eppendorf tube.

The MiSeq sequencing of the libraries was made at the Plateforme d’Analyses Génomiques (IBIS, Université Laval).

### 2.7. Data Filtering

All V4 sequences of the 16 rRNA gene were analyzed using the Divisive Amplicon Denoising Algorithm (DADA2) v1.20 pipeline (available at https://github.com/benjjneb/dada2 and accessed on 15 October 2022) as part of R V4.1.1. Sequences were cut and filtrated after their quality profiles with the function “filterandtrim” to remove poor-quality sequences. Error models were then constructed and the DADA2 algorithm was used to detect and treat errors before the merging of forward and reverse pairs. An ASV (amplicon sequence variant) table was made, and a step of chimera removal was performed with the function “removeBimeraDenovo”. Taxonomic annotation of the ASV was then carried out using the SILVA 138.1 database (last update, July 2020). The annotation was performed using the functions “assignTaxonomy” and “addSpecies” (package DADA2). A pre-trained classifier provided by the DADA2 developers (the file silva_nr99_v138.1_wSpecies_train_set.fa) was used with the functions “assignTaxonomy”. The annotated ASV, the metadata file, and the abundance data were grouped into a phyloseq object for analyses. ASV abundances were then normalized by using the function transform_sample_counts that divides the occurrence number of each ASV of a sample by the sum of all occurrences of all ASVs in the same sample. After the above-described filtering steps, ASV tables were suitable for downstream analyses. All samples were normalized before the alpha diversity indices were calculated so that the sum of the amplicon sequencing variants (ASVs) counts in each sample was equal to 1, allowing a relative comparison of microbial profiles between samples.

### 2.8. Abundance Tables of the Bacterial Genera of the 100 Most Abundant ASVs by Colony

Once the ASV table had been created, the 100 most abundant ASVs for each colony were grouped together in abundance tables at genus level, by experimental group, and sampling time (t0 or t21). The abundance of the main taxa was compared between experimental groups and between times within experimental groups using Tukey tests.

### 2.9. Alpha Diversity Indices

Alpha diversity indexes were used to quantify the bacterial community diversity. In this study, two indexes were calculated: Chao1 and Pielou, and one entropy measure was used as follows: Shannon. Here, the Chao1 index assesses ASV richness, which is the total number of ASVs, while the Pielou index assesses evenness, which is the distribution of ASV abundances, whereas the Shannon entropy considers both richness and evenness. The values of alpha diversity were compared between groups and times using the *t*-test.

### 2.10. Beta Diversity Indices

To examine potential differences between microbiota samples, a dissimilarity analysis was performed in R studio using the phyloseq file grouping the annotated ASV abundances and their treatment group. Two outliers were removed for this analysis due to an abnormal Bray–Curtis distance. The dissimilarity refers to the genetic difference or distance between bacterial communities. Two types of analysis were performed. Non-metric Multidimensional Scaling (NMDS) is a multivariate analysis used to visualize and compare communities according to their dissimilarity. It groups similar samples and keeps different samples apart. Principal Coordinate Analysis (PCoA) has the same objective as NMDS but seeks to find the principal axes that best explain the variation in the dissimilarity data and projects individual samples according to their dissimilarity values on these axes to visualize their relationships. Four measures of beta diversity were used in this study. The Bray–Curtis dissimilarity takes into consideration the presence and the relative abundance of the microbial species. Unifrac is a metric distance that measures the differences between communities, taking into account both the evolutionary relationships between taxa and their presence/absence, while the weighted Unifrac distance also takes into account the relative abundance of each ASV. PERMANOVA were carried out to compare the beta diversity between groups and times.

### 2.11. Interaction Networks

Taxonomic interaction networks were built to examine the co-occurrence and co-avoidance of ASVs inside each treatment group. These interaction networks were made in Cytoscape (v.3.10.1, Boston, MA, USA) with taxa and ASV tables generated in R studio using the phyloseq package. Each ASV was labeled at the genus level to reduce the number of unassigned sequences. Networks were constructed with the association inference tool CoNet 1.1.1.beta [35]. They were based on four different correlation measures: Pearson and Spearman’s rank correlations between ASVs, mutual information, and Bray–Curtis and Kullback–Leibler correlations. In total, 100 renormalized permutations and 100 bootstrap scores were generated for each measure and edge to mitigate compositionality bias. Only connections satisfying all four methods were retained. Statistical significance was evaluated using measure-specific *p*-values, which were computed and merged using Brown’s method [36]. To address multiple testing, the Benjamini–Hochberg procedure was applied [37], with a *p*-value threshold of 0.05. To visualize co-occurrence patterns as networks, the yFiles organic layout of Cytoscape was used. Network nodes represent ASVs while edges represent significant positive or negative correlations between these ASVs. Nodes and edges were color-coded to identify bacterial classes and positive or negative interactions, respectively. The size of a node reflects its connection strength. Statistical analyses were carried out by subdividing each group into three subgroups obtained by random selection to form sub-networks. By recording the parameters of these sub-networks on the same file, it was possible to compare the differences between groups using ANOVA (the data being normally distributed with homogeneous variances).

## 3. Results

### 3.1. Measurement of Zootechnical Performance Traits

To confirm the experimental groups’ homogeneity, ANOVA analysis was carried out on the weight and quantity of bees in each hive before treatments (Appendix A). These tests showed no difference between the groups at t0 (*p*-values 0.757 and 0.330, respectively).

The zootechnical performance traits are represented in Figure 2. Figure 2c–e show variables measured two days after wintering. Figure 2a,b,f represent variables measured a month after wintering (the dead colonies were removed prior to statistical analysis). There was no significant difference between groups for the number of uncapped brood cells (Figure 2b). Contrastingly, there were significant differences between groups for the number of capped cells (ANOVA, *p*-value = 0.022), as represented in Figure 2a. The capped cell number was significantly higher for the Bactocell^®^ group than for the Fumidil B^®^/Bactocell^®^ group (Fisher test; *p*-value = 0.004) and marginally higher compared to the control and Fumidil B^®^ (Fisher test *p*-value = 0.052, and 0.089, respectively). A similar tendency is observed for the total number of brood cells (Figure 2f), with significantly higher number of total brood cells in Bactocell ^®^ colonies compared to Fumidil B^®^/Bactocell^®^ colonies (Wilcoxon test, *p*-value = 0.01). Although the control group seems to have a smaller winter weight loss relative to all other groups (Figure 2c), there was no significant differences between groups. For the rate of bee loss during wintering (Figure 2d), there was no difference between groups (*p*-value = 0.189). Finally, for the number of bees per colony (Figure 2e), the Kruskal–Wallis test showed significant differences between groups (*p*-value = 0.037) and the Wilcoxon test indicates that the Bactocell^®^-treated colonies had marginally and significantly more bees than the control and the Fumidil B^®^/Bactocell^®^ colonies at 90 and 95 % confidence levels, respectively (*p*-value = 0.09 and 0.018 respectively). No difference was detected with the Kruskal–Wallis test for the *V. destructor* infestation rate (*p*-value 0.368). However, the Wilcoxon test showed a trend where the Fumidil B^®^ group had marginally more varroa than the control group (*p*-value 0.094).

### 3.2. Abundance Tables of Bacterial Genera of 100 Most Abundant ASVs by Colony

The 100 most abundant detected gut bacterial genera, according to treatment, are represented in Figure 3. For each group, the figure on the left shows the genera present in the 100 ASV most abundant at t0 (before treatment) and the figure on the right shows the genera abundance at the end of the three weeks of treatments (t21).
Time comparison:

In the four groups, bees sampled at t0 showed more taxa than the bees sampled after 21 days. The 100 most abundant ASVs represent 60% to 100% of the total number of sequences in all charts. The most abundant genus is *Lactobacillus*. For the control (Figure 3a) at t0, the genera *Arsenopoenus*, *Bartonella*, *Bifidobacterium*, *Bombilactobacillus*, *Commensalibacter*, *Gilliamella*, *Lactobacillus*, *Pseudomonas*, *Serratia*, and *Snodgrassella* are represented. At t21, *Pseudomonas* and *Serratia* decreased under the threshold of the 100 most abundant taxa. On the contrary, two new genera were detected at t21, *Frischella* and *Limnobacter*. The *Bartonella* genus appeared to be less abundant, while the abundance of *Snodgrassella* and *Gilliamella* seems to have increased. The Tukey test comparing the abundance of each genus between t0 and t21 showed a significant decrease of 3.8% for the genus *Commensalibacter* and a significant increase of 7.3% for the genus *Gilliamella* for the Control (*p*-values of 0.033 and 0.044, respectively). In the case of Bactocell^®^ (Figure 3b) at t0, *Arsenophonus*, *Bartonella*, *Bifidobacterium*, *Bombilactobacillus*, *Commensalibacter*, *Frischella*, *Gilliamella*, *Lactobacillus*, and *Snodgrassella* were the most abundant genera, as well as a non-identified genus. At t21, the genus *Arsenophenus* and the non-identified genus abundances dropped below the 100 most abundant taxa threshold. The Tukey test showed a significant decrease of 1.7% for the genera *Arsenophonus* in the Bactocell group between t0 and t21 (*p*-value 0.005). The Fumidil B^®^ group (Figure 3c) at t0 shared all but one genus with the Bactocell group (*Frischella*). The genus *Arsenophonus* and a non-identified genus became undetected at t21 whereas *Frischella* was detected. The Tukey test showed a marginally significant decrease of 2.6% for the genus *Commensalibacter* and significant decrease of 1.2% of *Arsenophonus* and a significant increase of 8% for *Gilliamella* for Fumidil B^®^ between t0 and t21 (*p*-values of 0.099, 0.013 and 0.004, respectively). Finally, for the bees that received the Bactocell^®^/Fumidil B^®^ combination (Figure 3d), *Arsenophonus*, *Bartonella*, *Bifidobacterium*, *Bombilactobacillus*, *Commensalibacter*, *Gilliamella*, *Lactobacillus*, *Methylobacterium-methylorubrum*, *Pantoea*, *Snodgrassella*, and an unidentified genus were detected at t0. At t21, eight of them were still detected: *Arsenophonus*, *Bartonella*, *Bifidobacterium*, *Bombilactobacillus*, *Commensalibacter*, *Gilliamella*, *Lactobacillus*, and *Snodgrassella*. In addition, *Frischella* and *Limnobacter* became detectable at t21. For this group, the Tukey tests showed a significant increase of 3.2% in *Lactobacillus* and a significant decrease of 5.3% in *Commensalibacter* between t0 and t21 (*p*-values of 0.038 and 0.069, respectively).
Groups comparison:

At t0 (Figure 3, graphs in the left), all groups presented similar taxonomic structure profiles despite little differences in the presence/absence of some genera (*Pseudomonas* and *Serratia* for three colonies of the control and *Methylobacterium-methylorubrum* and *Pantoea* for one hive of the Bactocell^®^/Fumidil B^®^ group). The Tukey tests used to compare the groups for each bacterial genus showed no significative differences between groups at t0, except for the Bactocell^®^/Fumidil B^®^ group that had significantly less bacteria of the genus *Lactobacillus* than Fumidil B^®^ at t0 (*p*-value 0.038).

At t21 (Figure 3, graphs in the right), the control and the Bactocell^®^/Fumidil B^®^ groups had similar taxonomic structure profiles with *Arsenophonus*, *Bartonella*, *Bifidobacterium*, *Bombilactobacillus*, *Commensalibacter*, *Gilliamella*, *Lactobacillus*, *Snodgrassella*, and *Limnobacter*. The two other groups showed the same genera but without *Arsenophonus* and *Limnobacter*. The Tukey tests used to compare the groups showed that the Bactocell^®^/Fumidil B^®^ group had significantly more *Bifidobacterium* than the Bactocell^®^ group (*p*-value 0.026) and marginally more *Bifidobacterium* than the control group (*p*-value 0.078) at t21. Moreover, at t21, the Bactocell^®^, FB, and Fumidil B^®^ groups had significantly more *Bombilactocillus* than the control group (*p*-value 0.000, 0.008, and 0.041, respectively).
Presence of Pediococcus genus bacteria in ASV tables:

A total of seven strains of *Pedioccocus* sp. bacteria (corresponding to the probiotic Bactocell^®^) were found in 14 of the colonies tested at t21 in the ASV tables. Of these colonies, six belonged to the Bactocell^®^ group and eight to the Fumagilin^®^/Bactocell^®^ group. No strain of this genera was detected in the control or Fumidil B^®^ groups, suggesting that no strains of *Pediococcus* sp. other than the probiotic Bactocell^®^ were naturally present in the honeybee colonies.

### 3.3. Alpha Diversity Indices

Comparison between groups before (t0) and after treatment (t21)

No significative difference in alpha diversity was observed at t0 and at t21 between the microbiota of the different groups (Appendix A).
Comparison between times

In Table 2, the alpha diversity of each group was compared between t0 and t21 using the Chao1, Shannon, and Pielou indexes. The *t*-test showed significant temporal differences in alpha diversity for each of the three indexes (*p*-values of 0.001 for Chao1, 0.001 for Shannon, and 0.003 for Pielou). For Chao1, the only significant difference concerned the Bactocell^®^/Fumidil B^®^ group, whose alpha diversity increased between t0 and t21. Significant temporal differences were observed for each of the groups with the Shannon and Pielou indices, with decreasing evenness for the control and Bactocell groups and increasing evenness for the Fumagilin group. The alpha diversity of the Bactocell^®^/Fumidil B^®^ group increased when evaluated with the Shannon index (as with Chao1) but decreased with the Pielou index. The Shannon index is both influenced by specific richness and evenness, showing the importance of other indexes such as Chao1 and Pielou to obtain independent measure of richness and evenness, respectively.

### 3.4. Beta Diversity Indices

The beta diversity indices are used to estimate dissimilarity between groups using both dissimilarity index and phylogenetic distance measures.
Comparison between groups before (t0) and after treatment (t21)

Appendix A represent the NMDS and PCoA analysis realized with the Bray–Curtis dissimilarity and the unweighted and weighted Unifrac distances before treatment (t0). The Permanova shows no difference between groups at t0. Appendix A represent the NMDS and PCoA analysis realized with the Bray–Curtis dissimilarity and the unweighted and weighted Unifrac distances and at t21. Again, the Permanova did not show any significant difference between groups at t21.
Comparison between times

Appendix A represent, respectively, the NMDS and PCOA analysis realized with Bray–Curtis dissimilarity and the unweighted and weighted Unifrac distances at t0 and t21. In all charts, the ellipses overlap and Permanova did not show any significant difference between groups.

### 3.5. Network Interactions

Network interactions of the bacterial genera of the microbiota have been regrouped in Figure 4.

On these networks, the Bactocell^®^ group presents a higher number of nodes and edges than the control, unlike the Fumidil B^®^ group that has a lower number of nodes and edges than the control. These parameters are regrouped in Table 3. The negative interactions rate in the Fumidil B^®^ group is also three times higher than in the control. Finally, the Bactocell^®^/Fumidil B^®^ group has a lower number of nodes but a similar number of edges and level of negative interactions than in the control.

Sub-networks were created to compare the groups at the end of treatment in terms of microbiota interaction networks. The parameters of these sub-networks have been grouped together in Appendix A and the means for each group in Table 4.

The ANOVA revealed a significative difference between groups for the rate of negative interactions (*p*-value 0.0387). The Bactocell^®^ group had a marginally lower rate of negative interactions than the control (*p*-value 0.058) and a significantly lower rate of negative interactions than Fumidil B^®^ (*p*-value 0.006). Similarly, the Fumidil B^®^/Bactocell^®^ group had a marginally lower rate of negative interactions than the Fumidil B^®^ group (*p*-value 0.098). However, no significant difference was detected in the variation in negative interactions between t0 and t21 (*p*-value 0.377), nor in the total number of interactions depending on the treatment (*p*-value 0.548), nor in the clustering coefficient (*p*-value 0.12), nor in the average number of neighbors (*p*-value 0.679).

## 4. Discussion

### 4.1. Measurement of Zootechnical Performance Traits

Post-wintering recovery of the colonies is a key factor in temperate and northern countries such as Canada, where honeybees suffer a long winter with limited resources (usually sugar syrup fed in autumn). Most colony loss occurs during this period [4,5]. The causes of this loss likely involve exposure to multiple stressors and their synergy [6]. It is therefore important to understand the factors responsible for such high mortality and to develop mitigating strategies to improve honeybees’ health status. In this study, several differences between groups can be highlighted regarding post-wintering recovery (Figure 2). The total number of bees was significantly higher for the colonies that received the probiotic than for the control colonies (Figure 2e). The number of capped brood cells was also significantly higher for the probiotic group than for the three other groups (Figure 2a). This tendency was observed for the total number of brood cells as well (Figure 2f). These three variables could be related, since a greater number of bees in the colony could lead to an increase in capping activity, or vice versa. The number of capped brood cells is the number of bees that are in the pupa stage, close to their adult form. A significant increase in this number had already been observed with the probiotic Bactocell^®^ in Bleau and coll. (2023) [38] where colonies were supplemented with Bactocell^®^ two times in autumn and two times in spring, but at a higher amount (1 L versus 50 mL in the present study) and concentration (10^9^ CFU versus 10^8^ CFU in the present study). Moreover, other probiotics of the lactic acid bacteria (LAB) type (to which the strain constituting Bactocell^®^ belongs) have already proved their worth in beekeeping [39]. In the study of Elenany and Hassan (2023) [40], endogenous LAB bacteria significantly improved the activity of honeybee colonies. Similarly, Budge and coll. (2016) [41] showed that LAB were correlated to a higher number of bees in colonies. Therefore, the probiotic Bactocell^®^ holds great promise as a treatment to help bees cope with the stress factors of wintering, and to give them a better start to the new season. The combination group had a lower performance than the other groups in most cases, although not significantly different from the control group. This trend may reflect an interaction between the probiotic and antibiotic in the syrup, even if these products were only mixed into the syrup one hour before supplementation. Another hypothesis could be that the traumatic dysbiosis generated by the antibiotic could have been aggravated by the response of some honeybee gut strains interacting with the probiotic. Further experiments in controlled environments could provide a better understanding of the mechanisms involved.

### 4.2. Abundance Tables of the Bacterial Genera of the 100 Most Abundant ASVs by Colony

At t0, the groups had no significative differences in terms of microbiota composition (Figure 3), except for the FB group that had less *Lactobacillus* than the Fumidil B^®^ group (*p*-value 0.026). Knowing the importance of this genus for microbiota homeostasis [42], this supports the hypothesis that the FB group may have been slightly weakened at the start of the experiment. At t21, the microbiota taxonomic composition of the FB group converged to that of the control group although it had marginally more *Bifidobacterium* than in the control and Fumidil B^®^ groups (Figure 3). Like *Lactobacillus*, the *Bifidobacterium* genus has genes encoding enzymes degrading complex plant sugars [14]. The low initial abundance of *Lactobacillus* may have allowed *Bifidobacterium* to proliferate due to decreased competition for resources. Nevertheless, the FB group showed a significant increase in *Lactobacillus* between t0 and t21, possibly due to the probiotic, although no change in abundance was detected for this genus for the Bactocell^®^ group between t0 and t21. All groups exhibited a significant decrease in *Commensalibacter* between t0 and t21, except for the Bactocell^®^ group. This genus, as well as *Lactobacillus*, has been associated with the microbiota of thriving hives [19]. In contrast, the genus *Gilliamella* is known to increase in non-thriving colonies [19] and increased significantly in the control and Fumidil B^®^ groups. For all groups except Bactocell^®^, the genus *Frishella* also increased at t21. *Frishella perrara* produces secretions creating antagonist interactions with other bacteria and activating the host immune system [15]. Finally, the Bactocell^®^ and Fumidil B^®^ treatments led to a significant decrease in the genus *Arsenophonus* between t0 and t21. This reduction could be linked to a rebalancing of the microbiota leading to a reduction in rare species in the Bactocell^®^ group, and to the antimicrobial activity of Fumidil B^®^. *Arsenophonus* (Gammaproteobacteria, Enterobacteriales) was found in all groups at t0 and in the control and FB groups at t21. Previously detected in honeybee samples [43], it is still understudied but its prevalence was observed to increase with symptoms of colony collapse disorder [44] and is linked to poor honeybee health [41]. Although not assigned here, a strain of this genus has been identified as responsible for son-killer disease in the parasitic wasp *Nasonia vitripenni* [45]. Other pathogenic taxa such as *Serratia marcescens*, *Hafnia alvei*, *Escherichia coli*, *Melissococcus plutonius*, and *Paenibacillus larvae* were detected at low abundance. *H. alvei* was detected in two control group colonies at t21, a group that also contained *S. marcescens* in three of its colonies. *E. Coli* was detected in the FB group at t21. At t0, two FB group colonies also contained unidentified strains from the *Peanibacillus* genus, one of whose species, *Peanibacillus larvae*, is known to cause American foulbrood. One of these two FB colonies died during winter, likely from another cause, as no visible symptoms of American foulbrood were observed. In addition, *Peanibacillus* was no longer detected at t21 in the second FB colony. The presence of these pathogenic and putative pathogenic strains could also have had an impact on the health of the concerned colonies, particularly in the FB group.

### 4.3. Alpha Diversity Indices

The comparison of alpha diversity with Chao1 and Pielou did not show differences at t0 nor at t21 between groups (Appendix A but showed significant differences within the groups between t0 and t21 (Table 2). The FB group encountered a significant increase in species richness (Chao1). This index is influenced by rare species (i.e., species that do not belong to the core microbiota) like the genus *Limnobacter* (neutral), and non-abundant and poorly connected taxa such as *E. coli* or *Peanibacillus* (both being potentially pathogenic) [46]. Chao1 interpretation must be complemented with other indicators. Evenness decreased for the control, Bactocell^®,^ and FB groups, whereas it increased for Fumidil B^®^. The decrease in evenness observed for Bactocell^®^ indicates a less even distribution of species abundances. This could reflect a more structured and balanced microbiota where the rare taxa decreased (*Arsenophonus*, *Enterobacter*, for example) while the core taxa increased or remained unchanged (*Commensalibacter*, *Lactobacillus*, *Bombilactobacillus*). Contrastingly, the increasing evenness indicates a more even distribution of species abundances, which might reflect a dysbiosis in the microbiota following the Fumidil B^®^ exposure since core species decreased in abundance (*Commensalibacter*) while rare taxa increased (*Frischella*, *Methylobacterium-Methylorubrum*, and *Pantoea* genera, Protobacteraceae family, and possibly other low abundance taxa). In the FB group, the species richness increased (Chao1) whereas the evenness decreased (Pielou). On the one hand, these results could come from the rise in rare strains, increasing species richness; on the other, the decrease in Pielou indicates that dominant strains at t0 did not decrease in abundance at t21 (*Bartonella*, *Bombilactobacillus*, *Giliamella*, *Snodgrassella*) or increased at t21 (*Lactobacillus*, *Bifidobacterium*) as in the Fumidil B group. This pattern suggests that combining Bactocell^®^ and Fumidil B^®^ mitigates the gut dysbiosis triggered by Fumidil B^®^ by restoring key species of the microbiota [14,42].

### 4.4. Beta Diversity Indices

The various beta diversity indices used to assess the dissimilarity of the colony microbiota showed no differences between groups at t0 and t21 (Appendix A) nor within each group between t0 and t21 (Appendix A). Therefore, the treatments did not induce noticeable strain replacement between groups and/or after treatment, despite significant differences in evenness and zootechnical traits between treatments. Rather, these differences co-occurred with changes in microbiota composition in terms of species richness and evenness. A few alternative hypotheses could explain these observations. Firstly, a dilution effect could have occurred since this test was performed at the colony scale, resulting in mitigated effects on microbiota beta diversity. Although less likely, this scale could have prevented the homogeneous distribution of the product between individuals. Nevertheless, the genus *Pediococcus* (to which the Bactocell^®^ strain belongs) has been detected in over 50% of the colonies treated with Bactocell^®^ (and not at all in colonies from other groups). This genus was detected at a very low abundance and may simply be below the detection limit for other probiotic-treated colonies. The impact of treatments may also not be apparent for beta diversity at the scale of colonies in the same environment. Rather, it appeared that the experimental treatments induced changes in both the alpha diversity (species richness and evenness, as stated above) and interaction networks in which were observed significant differences between groups.

### 4.5. Network Interactions

Network interactions were based on pairwise co-occurrence patterns assessed from 16S rRNA gene amplicon sequences, interpreted as taxonomic interactions from a microbial ecology perspective. When an increase in abundance of one strain is associated with an increase in abundance of another, it indicates a positive interaction. Conversely, if such an increase corresponds with a decrease in abundance of the other, the interaction is negative. If there is no correlation between the abundance variations in the two taxa, the interaction is considered neutral and is not represented in the network. The biological meaning of positive interactions can be metabolic complementary (i.e., syntrophy), for example. Likewise, a negative interaction could result from competition. Therefore, an increased number of positive interactions is interpreted as a hallmark of increased stability and connectivity of the microbiota community. High connectivity combined with a high rate of positive interactions confers greater resilience to microbiota, since a key function could be performed by several taxa (i.e., redundancy) and thus secured in case of the disappearance of one taxon [47,48]. On the other hand, an increased number of negative interactions is interpreted as a hallmark of dysbiosis [12,49,50]. Based on this framework, it is possible to compare these networks and to define which one is bonded to a better microbiota and honeybee health (Table 3).

As expected, all groups were homogenous at t0, but the group receiving the probiotic/antimicrobial (FB) combination had the least dense network. As observed on the alpha diversity, this rather weak initial state of the microbiota for this group at t0 could explain its poor performance for spring recovery. At t21, statistical analyses were carried out in sub-networks to highlight any significant differences between groups in terms of negative interactions. The Bactocell^®^ group had the lowest number of negative interactions (a parameter frequently associated with dysbiosis [50]), significantly lower than the Fumidil B^®^ group and marginally lower than the control group. The Bactocell^®^ group also had the higher number of edges at t21 (but not significative). This strongly suggests a beneficial effect of Bactocell^®^ on the interaction network, especially as this group had the highest rate of negative interactions at t0, before probiotic administration. This probiotic improves the stability of the interaction network within the microbiota, in line with the alpha diversity results, and could therefore be linked to the better resiliency after wintering recorded for this group. Since the best post-wintering recovery was mostly observed for the number of capped brood cells, probiotic supplementation could also benefit the queen or brood. On the other hand, the Fumidil B^®^ group’s associated microbiota showed an unbalanced interaction network. For pooled networks, the number of negative interactions tripled for the Fumidil B^®^ group compared to the control (Table 3) while it had the fewest negative interactions before treatment (t0). The decrease in the total interaction rate with Fumidil B^®^ could be explained by the elimination of key specific taxa, inducing a lack of resources. In turn, this lack of resources could have increased competition between bacterial populations, leading to more negative interactions. These interaction shifts can be interpreted as dysbiosis signature, i.e., the sign of an impaired microbiota [12]. Considering that bees feed almost exclusively on syrup after autumn feeding, it is likely that a microbiota disturbed by an antimicrobial (such as Fumidil B^®^, used at the end-of-season feeding) will have difficulty returning to a homeostasis state. Overall, significant impairments in the honeybee microbiota have been repeatedly associated with the use of antibiotics such as tetracycline-derived compounds [7,51]. Collectively, these studies highlight that frequently used antibiotics reduce both the abundance and genetic diversity of microbiota strains. Finally, the interaction network of the FB group looked like the control group. This supports the hypothesis that Bactocell^®^ compensates for the negative effect of the antimicrobial compound when given repeatedly with and after this treatment. Interestingly, the FB group showed marginally fewer negative interactions than the Fumidil B^®^ group. A recent study also showed that Bactocell^®^ restored the bee microbiota to a state close to the control after the induction of dysbiosis by a stressor factor (*V. ceranae*) [52]. In this context, Bactocell^®^ could be administrated as a complementary product to mitigate the negative impacts exerted by Fumidil^®^, and possibly other antimicrobials, on bee microbiota, especially prior to wintering.

All groups showed an increased number of negative interactions at t21. However, while this increase was negligible (+0.6%) in the control group, negative interactions were multiplied by 8 in the Fumidil B^®^ group. The sub-networks showed a similar trend (Table 4), the difference being significant in the Bactocell^®^ group and marginally significant in the FB group. Although not significant, the average number of neighbors decreased in all groups except Bactocell^®^. Moreover, the two groups that did not receive Bactocell^®^, the control and Fumidil B^®^, had a lower number of edges, whereas the two groups that received Bactocell^®^ harbored denser interaction networks at t21. The Bactocell^®^ group exhibited the highest number of edges. These results may suggest that administration of Bactocell^®^ actually alleviates the stress factors associated with wintering.

The bacterial microbiota core members are the main ones to be positively connected to each other. *Lactobacillus* and *Bombilactobacillus* were the most connected in all experimental groups, at both t0 and t21. This conservative interaction pattern highlights the importance of these core members, known to improve the microbiota homeostasis by supplying easily metabolizable molecules, thus decreasing interspecific competition [42,53]. Rarer genera, such as *Arsenophonus*, were also present at t0 and sometimes negatively correlated with core members, suggesting a possible pathogenic effect. At t21, rare taxa disappeared from all networks except the control. This shows the efficiency of Bactocell^®^, Fumidil B^®,^ and their combination to suppress interactions of rare and potentially pathogenic taxa with core members of honeybee microbiota. Furthermore, the alpha diversity data showed that the Bactocell^®^ group exhibited reduced evenness, accompanied by a decrease in rare taxa (e.g., *Arsenophonus*, *Enterobacter*), while some core taxa abundance remained stable or increased, suggesting an antagonistic activity of the probiotic against bee opportunistic pathogens (e.g., *Paenibacillus* genus), as observed in the literature [54,55]. Overall, our results suggest that network analysis is a promising approach to leverage 16S rRNA gene metabarcoding datasets as biomarkers of the bacterial community integrity in untargeted analyses of contamination and to better understand how specific stressors alter microbiota homeostasis.

To conclude on the biological significance of the main results generated by the present study, we would like to recall that the changes detected in the composition of the microbiota can have positive or negative impacts on bee health at the individual level. These compositional changes can either prevent (e.g., increased colonization resistance) or favor (e.g., decreased colonization resistance) the establishment of opportunistic and potentially pathogenic taxa. Moreover, they can promote or alter certain functions essential to bees that depend on particular taxa. Thus, by affecting bee health, variations in microbiota composition also influence the productivity of their colonies, as healthier bees are more capable of fully performing the colony’s main tasks.

### 4.6. Limits and Perspectives

Carrying out a survey in apiaries renders the control of environmental parameters very difficult. Therefore, by generating a non-neglectable background noise, it was difficult to detect significant differences between experimental groups, despite the sample size per group being in the upper range of the available references. Among uncontrolled events, looting phenomena may have occurred between colonies, with a possible slight dispersion of the products tested across experimental groups. In the future, it may be appropriate to optimize the doses or methods of probiotic supplementation (mixing with pollen patty or providing continuous supplementation during wintering, for example). In addition, it would have been interesting to carry out transcriptomic analysis to assess the functional impact of the tested products. Similar tests carried out in a controlled environment (i.e., caged bees) could also be relevant for understanding the underlying mechanisms associated with the products tested. This study represents a first step towards helping honeybees adapt to exposure to chemical molecules and hindering the multiplication of hive pathogens, with a probiotic solution that is applicable and available to beekeepers. It also highlights the key role played by the intestinal microbiota in bee survival and colony performance. Finally, it paves the way for research into a global solution to improve bee health in anticipation of exposure to multiple stressors.

## 5. Conclusions

This study shows a beneficial effect of the probiotic Bactocell^®^ at the colony scale: it favors the increase in both adult bee population and the number of capped brood cells in spring. It could be administrated as a food supplement before wintering, to strengthen the colonies in spring and kick-start the new apiary season. In this case, further tests will be needed to improve the dose, method, and schedule of administration. Bactocell^®^ is also correlated with a more structured microbiota (alpha diversity and interaction networks) where the core taxa members thrive, supplying nutritive metabolites to the bee that also promote detoxification [42,53]. It structures the bee microbiota and makes it more resilient, suppressing most signs of dysbiosis detected at t0. This treatment is also correlated with a decrease in rare and potentially pathogenic taxa. This was observed for Fumidil B^®^ as well, although the antimicrobial decreased core bacteria members too, leading to obvious microbiota disbalance highlighted with an increase in negative interactions. As observed for other antibiotics, Fumidil B^®^ therefore induces dysbiosis for the bees, even at the colony scale, although no effects were detected relative to hive productivity. Finally, the addition of Bactocell^®^ to Fumidil B^®^ seems to have mitigated the toxicity of the antibiotic compound since the microbiota taxonomic structure converged with that of the control group. Nevertheless, the colony productivity of this group showed a tendency to be slightly lower than the other groups, possibly due to a weakened microbiota at t0. Comparison with the initial state of the control network shows that all groups may have suffered a background deterioration in their microbiota due to a decrease in temperatures. However, the probiotic groups (Bactocell^®^ and FB) presented a better density after treatment and Bactocell^®^ could be proposed to restore bee health during/after antimicrobial treatment or/and in the event of thermic stress.

## Figures and Tables

**Figure 1 microorganisms-12-01154-f001:**
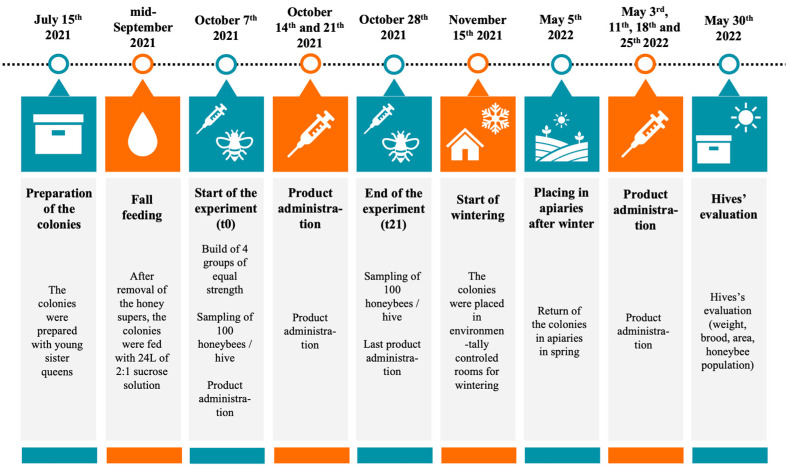
Chronology of the experiment. Colonies were prepared in July 2021. Honey supers were removed in early September 2021 and feeding (sucrose solution 2:1) was implemented in mid-September. Exposure to study products began on 7 October (t0, day of first sampling) and was repeated weekly until 28 October (t21, second sampling). The colonies were then placed in an environmentally controlled room for the duration of the wintering period on 15 November. They were assessed at the end of the overwintering period (20 April 2022) for weight, brood, and honeybee population, and then placed in two apiaries.

**Figure 2 microorganisms-12-01154-f002:**
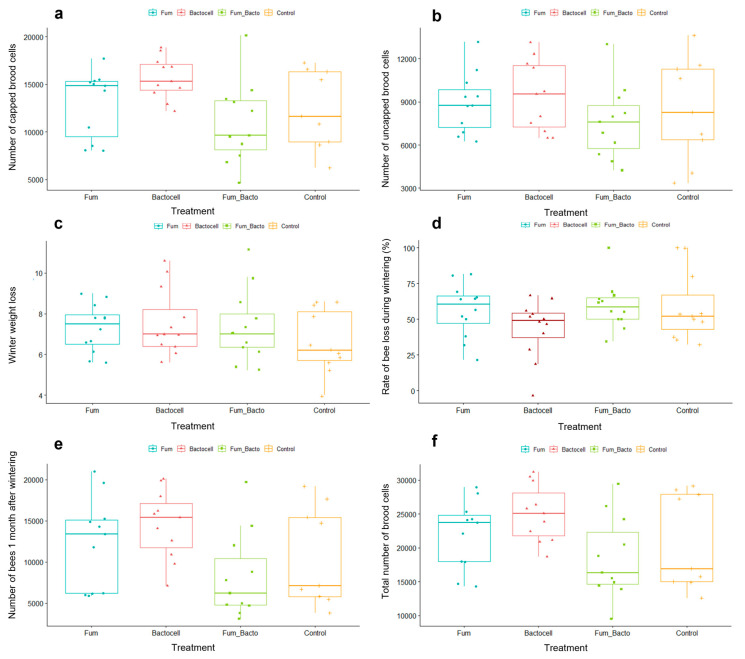
Zootechnical performance traits of colonies after overwintering. The (**c**,**d**) show variables measured two days after wintering. The (**a**,**b**,**e**,**f**) represent variables measured a month after wintering (the dead colonies were removed in statistical analysis). (**a**) Number of capped brood cells according to of treatment. Variable indicating future bees to be born in the colony. (**b**) Number of uncapped brood cells according to treatment. Variable representative of the queen’s egg-laying activity. (**c**) Winter weight loss. (**d**) Rate of bee loss during wintering (in %). (**e**) Total number of bees 1 month after wintering. Variable indicating colony strength. (**f**). Total number of brood cells by treatment. Variable indicating colony development.

**Figure 3 microorganisms-12-01154-f003:**
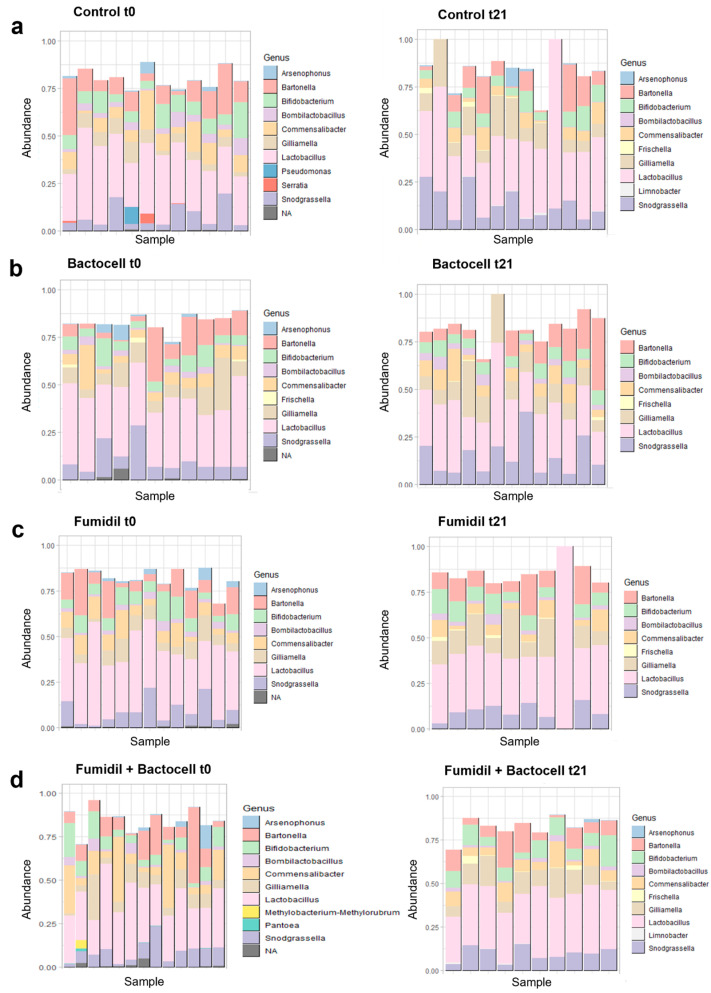
Representation of the bacterial genera present in the 100 most abundant ASVs in each colony, by group and sampling time (t0 and t21). For a minimal number of unassigned taxa, these ASVs are regrouped at the genus scale. The samples that do not reach 1.00% in abundance have rare taxa that are not represented in this figure. (**a**) Control group, at t0 on the left (before exposures) and at t21 on the right (after 3 weeks of exposure). (**b**) Bactocell^®^ group, at t0 and t21. (**c**) Fumidil^®^ group, at t0 and t21. (**d**) Fumidil^®^ + Bactocell^®^ group, at t0 and t21.

**Figure 4 microorganisms-12-01154-f004:**
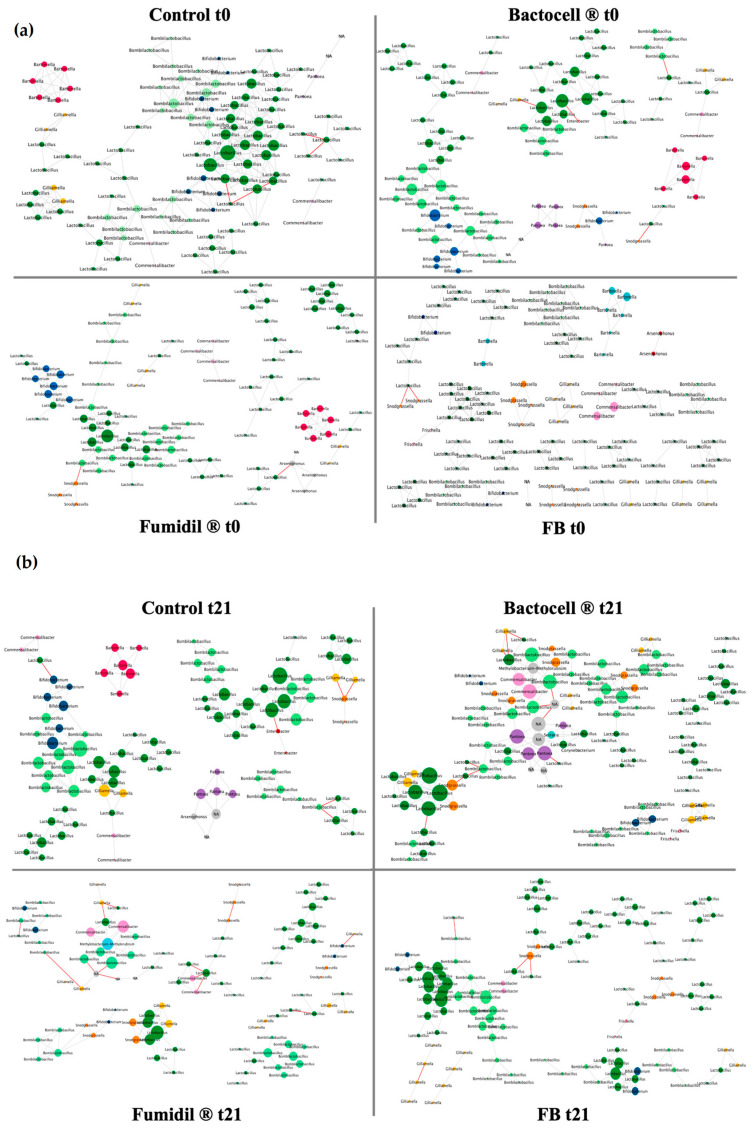
(**a**) Network interactions of the microbiota of the bees sampled from the groups at t0 (before treatment) and t21 (after treatment). The networks are based on the co-occurrence and the co-avoidance of ASVs. Taxa are figured under the form of a circle (node) at the scale of the genus. Each genus has a peculiar color. The size of a node is proportional to its connectivity. Lines represent interactions between taxa. A green line represents a positive relationship (the presence of one of the two linked taxa is positively correlated with the presence of the other), while red lines represent negative relationships. The control group is used as a baseline. (**b**) Network interaction of the microbiota of the bees sampled from the control group at t21 (after treatment). The networks are based on the co-occurrence and the co-avoidance of ASVs. Taxa are figured under the form of a circle (node) at the scale of the genus. Each genus has a peculiar color. The size of a node is proportional to its connectivity. Lines represent interactions between taxa. A green line represents a positive relationship (the presence of one of the two linked taxa is positively correlated with the presence of the other), while red lines represent negative relationships. The control group is used as a baseline.

**Table 1 microorganisms-12-01154-t001:** Experimental design. Each group was composed of 12 colonies that all received 50 mL of control syrup (mixed or not with the products) that was sprayed between the hive frames.

Group	Number of Hives	Treatment Week 1	Treatment Weeks 2, 3 and 4	Concentration (/mL)
Control	12	Control (syrup 1:1)	syrup 1:1	NA
Bactocell^®^	12	Bactocell^®^	Bactocell^®^	10^8^ CFU
Fumidil B^®^	12	Fumidil B^®^	syrup 1:1	1.25 × 10^−3^ g
FB	12	Fumidil B^®^ + Bactocell^®^	Bactocell^®^	10^8^ CFU + 1.25 × 10^−3^ g

**Table 2 microorganisms-12-01154-t002:** Table showing the average of alpha diversity indices (in order: Chao1, Shannon, and Pielou) according to groups, before (t0) and after (t21) treatments, and their variation between these two times as well as the *p*-value associated with this variation. As the data were normally distributed with homogeneous variances, *t*-tests for pairwise samples were carried out for each index, followed by Wilcoxon tests.

Treatment	Mean Chao1_t0	Mean Chao1_t21	Differences Chao1	*p*-Value Chao1
Control	278	252	−26	0.214
Bactocell	286	345	59	0.897
Fumidil B^®^	295	292	−3	0.300
Bactocell + Fumidil B^®^	286	374	88	0.011

Treatment	Mean Shannon_t0	Mean Shannon_t21	Differences Shannon	*p*-value Shannon
Control	4.51	3.86	−0.65	0.002
Bactocell	4.56	4.28	−0.28	0.024
Fumidil B^®^	4.49	4.59	0.1	0.001
Bactocell + Fumidil B^®^	4.24	4.47	0.23	0.026

Treatment	Mean Pielou_t0	Mean Pielou_t21	Differences Pielou	*p*-value Pielou
Control	0.809	0.78	−0.029	0.020
Bactocell	0.812	0.801	−0.011	0.003
Fumidil B^®^	0.797	0.817	0.02	0.000
Bactocell + Fumidil B^®^	0.792	0.781	−0.011	0.041

**Table 3 microorganisms-12-01154-t003:** Parameters of microbiota interaction networks for each group at t0 and t21 (number of nodes (i.e., number of bacterial taxa), number of edges (i.e., number of interactions), rate of negative interactions, variation in the negative interaction rate between t0 and t21, average number of neighbors for one node, and clustering coefficient.

Treatment	Variation in Number of Nodes between t0 and t21	Variation in Number of Edges between t0 and t21	Variation in n.i between t0 and t21	Variation in Average Number of Neighbors between t0 and t21	Variation in Clustering Coefficient between t0 and t21
Control	+18	−45	×1.4	−1068	+0.036
Bactocell^®^	+69	+91	×1.76	+460	−0.072
Fumidil B^®^	−17	−17	×8.12	−444	−0.177
FB	+24	+74	×2.12	−1205	−0.084

**Table 4 microorganisms-12-01154-t004:** Mean of the parameters of the sub-networks for each group and at t21. These sub-networks were obtained by subdividing each group into three subgroups obtained by random selection.

	Co-Occurrence	Mutual Exclusion	Total Interactions	Negative Interactions Rate	Variation in Negative Interactions
Control	619	311.33	930.33	33.46	−0.35
Bactocell^®^	681	279.67	960.67	29.11	−6.50
Fumidil B^®^	567.33	320	887.33	36.06	3.57
FB	599.33	288.67	888	32.51	−3.91

## Data Availability

The data are available in the NCBI Genbank^®^ with the accession number PRJNA1103710 (https://www.ncbi.nlm.nih.gov/nuccore/?term=PRJNA1103710, accessed on 1 May 2024).

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
