# Peer review of "Managing Microbiota Activity of Apis mellifera with Probiotic (Bactocell®) and Antimicrobial (Fumidil B®) Treatments: Effects on Spring Colony Strength"

_microorganisms, 2024, doi:10.3390/microorganisms12061154_

Round 1

Reviewer 1 Report

Comments and Suggestions for Authors

The present work presents an interesting study regarding the treatment of bees with antibiotics and probiotics, as well as their combined use. An important strength of this article is the relatively large number of colonies on which the study was done.

The study describes that the antimicrobial Fumidil B was only applied during the first of four treatment sessions, while the subsequent sessions differed between groups. Could the authors explain the rationale behind this design choice?

In the end, how many hives were used in experiment 50? you mentioned that you had 4 groups of 12 hives each…

Line 140 The administered syrup contained 108 CFU Bactocel?  or 108 CFU Bactocel? (Table 1)

In the methodology section, there is no mention of controlling for genetic variability among the bee colonies used in the study, it is known that some colonies might be naturally more resilient due to their genetic background, which could skew the effects of treatments like probiotics or antimicrobials.  Could you clarify if any measures were taken to assess or control for genetic differences between the colonies? Do you think genetic variability might have influenced the outcomes of your treatments?

While the controlled environment is a strength for reducing variability, it might also limit the ecological validity of the findings. Some treatments performed under field conditions, where environmental stressors and natural variability play a significant role, might differ from the presened results.

The authors notes that colonies treated with the combination of Fumidil B and Bactocell showed lower performance in some metrics compared to those treated solely with Bactocell. Could you discuss why the addition of Fumidil B appears to diminish the beneficial effects observed with Bactocell® alone? What interactions between these two treatments might be responsible for this observed phenomenon?

The results show that the combined treatment group (FB) did not perform as well as the Bactocell® alone group in terms of post-wintering recovery. Could you discuss the possible reasons for this observation? How might the interactions between the probiotic and antimicrobial agents in the combined treatment be affecting the microbiota and, consequently, the health of the bee colonies?

The conclusions support those presented in the text, but in my opinion this section should be shorter.

As a recommendation, it would be beneficial to follow the colonies over multiple seasons to assess the long-term effects of the treatments on colony health and survival rates. Conducting similar experiments in field conditions could help validate the findings and ensure they are applicable in real-world beekeeping scenarios.

Author Response

Q#1) The study describes that the antimicrobial Fumidil B was only applied during the first of four treatment sessions, while the subsequent sessions differed between groups. Could the authors explain the rationale behind this design choice? 

Thank you for this interesting question, the Fumidil B was only given once and during the first week of the experiment according to veterinary and product recommendation in order to reflect its current use in apiaries as added line 177 (“according to veterinary and product recommendation”).

Q#2) In the end, how many hives were used in experiment 50? you mentioned that you had 4 groups of 12 hives each…

Thank you for this comment, 48 colonies in total were used for this experiment, as corrected line 145.

Q#3) Line 140 The administered syrup contained 108 CFU Bactocel?  or 108 CFU Bactocel? (Table 1)

It was 108 CFU Bactocell. All occurrences have been corrected in the Table 1 and the text (lines 153,154 and 155).

Q#4) In the methodology section, there is no mention of controlling for genetic variability among the bee colonies used in the study, it is known that some colonies might be naturally more resilient due to their genetic background, which could skew the effects of treatments like probiotics or antimicrobials.  Could you clarify if any measures were taken to assess or control for genetic differences between the colonies? Do you think genetic variability might have influenced the outcomes of your treatments? 

This interesting concern have been raised during the preparation of the study and therefore we used young sister queens to develop the colonies to control this parameter (as mentioned line 117). Although they show slight genetic differences due to the different fathers, this measure should nevertheless have limited the effects of genetics in this experiment.

Q#5) While the controlled environment is a strength for reducing variability, it might also limit the ecological validity of the findings. Some treatments performed under field conditions, where environmental stressors and natural variability play a significant role, might differ from the presented results.

Thank you for your comment. As you rightly mentioned, field-scale experiments are associated with great variability in the results and observations obtained. However, a study also carried out in apiaries and with the same probiotic gave results similar to ours (increase in the number of capped cells) (see Bleau et al., 2023). Other studies also carried out with probiotics such as lactic acid bacteria have shown similar results. These studies are mentioned in the discussion in the lines 549 to 557: “A significant increase of this number had already been observed with the probiotic Bactocell® in Bleau and coll. (2023) [38] where colonies were supplemented with Bactocell® two times in autumn and two times in spring, but at higher amount (1L versus 50 mL here) and concentration (109 CFU versus 108 CFU here) compared to the present study. Moreover, other probiotics of the lactic acid bacteria (LAB) type (to which belongs the strain constituting Bactocell®) have already proved their worth in beekeeping [39]. In the study of Elenany and Hassan (2023) [40], endogenous LAB bacteria have significantly improved the activity of honey bee colonies. Similarly, Budge and coll. (2016) [41] showed that LAB are correlated to a higher number of bees in colonies.”.

Q#6) The results show that the combined treatment group (FB) did not perform as well as the Bactocell® alone group in terms of post-wintering recovery. Could you discuss the possible reasons for this observation? How might the interactions between the probiotic and antimicrobial agents in the combined treatment be affecting the microbiota and, consequently, the health of the bee colonies?

Thank you for your interesting question. The lower performance of this group is a tendency that was not significant for none of the six parameters tested. As mentioned in the discussion, this group might have been slightly weakened before the experiment (see lines 568 to 569 and 666 to 667). A little paragraph have been added in lines 561 to 567 to address this concern: “This trend may reflect an interaction between probiotic and antibiotic in the syrup, even if these products were only mixed into the syrup one hour before supplementation. Another hypothesis could be that the traumatic dysbiosis generated by the antibiotic could have been aggravated by the response of some honey bee gut strains interacting with the probiotic. Further experiments in controlled environments could provide a better understanding of the mechanisms involved.”.

Q#7) The conclusions support those presented in the text, but in my opinion this section should be shorter.

Thank you for this suggestion, we did our best to shorten conclusion, according to all reviewer’s comments.

Q#8) As a recommendation, it would be beneficial to follow the colonies over multiple seasons to assess the long-term effects of the treatments on colony health and survival rates. Conducting similar experiments in field conditions could help validate the findings and ensure they are applicable in real-world beekeeping scenarios.

Thank you for your recommendation. A multi-year study is currently being carried out to assess the long-term effects of these treatments.

Reviewer 2 Report

Comments and Suggestions for Authors

1.       What are the future implications of this study? Please discuss and justify.

2.       Are there any statistics used in this work? Authors need to include a separate section about statistical analysis.

3.       The authors need to discuss the limitations associated with the current study.

Author Response

REVIEWER 2

Q#1) What are the future implications of this study? Please discuss and justify.

Thank you for this relevant question. We added a new subsection at the end of the discussion (line 714) describing the perspectives (see lines 743 to 748: “This study represents a first step towards helping honey bees adapt to exposure to chemical molecules and hindering the multiplication of hive pathogens, with a probiotic solution that is applicable and available to beekeepers. It also highlights the key role played by the intestinal microbiota in bee survival and colony performance. Finally, it paves the way for research into a global solution to improve bee health in anticipation of exposure to multiple stressors.”).

Q#2) Are there any statistics used in this work? Authors need to include a separate section about statistical analysis.

Thank you for this comment. The statistical parts of this study are described in the material and methods, paragraphs 2.7 to 2.11. To address your concern, we added more information in these parts in the lines 227 (“These parameters were compared between groups using ANOVA or Kruskal-Wallis analysis (depending on their conditions of normality and homogeneity of variances).”, 292 (“The abundance of the main taxa was compared between groups and between times within the groups using Tuckey tests.”), 300 (“The values of alpha diversity were compared between groups and times by t.test.”), 316 (“PERMANOVA were used to compare the beta-diversity between groups and times.”) and 338 (“By recording the parameters of these sub-networks on the same file, it was possible to compare the differences between groups using ANOVA (the data being normal and their variances homogeneous).”).

Q#3) The authors need to discuss the limitations associated with the current study.

Thank you for this comment. As a result, we added a new subsection at the end of the discussion (lines 732-743) that describes the limits of the study: “Carrying out a survey in apiaries renders the control of environmental parameters very difficult. Therefore, by generating a non-neglectable background noise, it was difficult to detect significant differences between experimental groups, despite sample size per group was in the upper range of available references. Among uncontrolled events, looting phenomena may have occurred between colonies, with a possible slight dispersion of the products tested across experimental groups. In the future, it may be appropriate to optimize the doses or methods of probiotic supplementation (mixing it with pollen patty or providing continuous supplementation during wintering, for example). In addition, it would have been interesting to carry out transcriptomic analysis to assess the functional impact of the tested products. Similar tests carried out in a controlled environment (i.e. cage-bees) could also be relevant for understanding the underlying mechanisms associated with the products tested.”.

Reviewer 3 Report

Comments and Suggestions for Authors

This study investigates the impact of probiotic (Bactocell) and antimicrobial (Fumidil B) treatments on the microbiome dynamics of honey bee colonies post-wintering, offering insights into potential strategies to enhance colony health and productivity. While the manuscript presents intriguing findings regarding changes in microbial diversity, interactions, and zootechnical performance traits across treatment groups, several methodological and interpretational aspects require further clarification and validation to strengthen the scientific rigor and biological relevance of the study's conclusions.

Some comments to be addressed:

·      What is the relation between colony health and the gut microbiota of honeybees?

·      In the abstract, authors are required to show the effect of the combination of Bactocell and Fumidil on the colony health.

·      In the introduction line 34, the authors are advised to give more examples of these biotic and abiotic stressors.

·      Can you give more details about the type of probiotic bacteria present in Bactocell and whether it is only 1 type of bacteria or more than one type?

·      In line 45, what is meant by firm 4 and firm 5?

·      In lines 69 and 70, based on references 14 to 16, what is the new added value this research will highlight?

·      In lines 97 to 102, this part is a conclusion and should not be here in the introduction part.

·      The ethical approval number is missed.

·      In line 107, please include the year 2021 after July 5th. Also, this date is different from that mentioned in Figure 1. It is the 15th of July 2021 in Figure 1. Please double-check the correct one.

·      In lines 141 – 143, the numbers containing 10 to the power need to be adjusted.

·      In line 227, please mention the manufacturer country after the company name.

·      In line 256, the authors mentioned that they used SILVA Database, but which version did they use, and did the authors make any training for classifiers?

·      P value, the P letter should be italicized all over the manuscript.

·      Many other data could be grasped from the sequencing microbiome data which should be useful like ANCOM or LEFSE. Moreover, the authors didn't perform PICRUST to predict metabolic pathways differences between groups.

·      In line 416, the wrong spelling of Pediococcus sp. should be corrected.

·      What about microbiome data deposition and availability?

·      The results section is dense and lacks clarity in presenting key findings. The authors should consider reorganizing the results to improve readability.

·      The interpretation of results is often ambiguous, with the authors failing to clearly link findings to underlying biological mechanisms or theoretical frameworks. 

·      The manuscript contains numerous typographical errors and grammatical mistakes that detract from its overall professionalism. A thorough proofreading and editing process is needed to correct these issues.

·      How do the authors ensure the robustness of their experimental design, particularly regarding sample size and randomization procedures?

·      To what extent do the findings of this study align with or diverge from prior research on similar topics? Discuss whether the observed shifts in microbial diversity are consistent with expectations based on known ecological principles and previous research findings in honey bee microbiome studies.

·      How do the authors interpret the observed changes in alpha and beta diversity indices within and between groups?

·      What are the potential implications of the study's findings for honey bee health management?

·      Can the authors clarify the biological mechanisms underlying the observed interactions within the microbiota networks?

·      Are there any additional analyses or experiments that the authors believe would strengthen the study's conclusions?

·      Address the potential functional roles of the microbial communities identified, considering their taxonomic composition and known metabolic capabilities. Discuss any functional predictions or hypotheses derived from the taxonomic data.

·      Provide detailed information regarding the exact sampling protocol followed, including the specific anatomical sites sampled from honey bees, the sampling frequency, and any precautions taken to minimize environmental contamination during sampling.

·      Discuss how changes in microbial composition may impact honey bee health and colony productivity over time.

Comments on the Quality of English Language

NA

Author Response

This study investigates the impact of probiotic (Bactocell) and antimicrobial (Fumidil B) treatments on the microbiome dynamics of honey bee colonies post-wintering, offering insights into potential strategies to enhance colony health and productivity. While the manuscript presents intriguing findings regarding changes in microbial diversity, interactions, and zootechnical performance traits across treatment groups, several methodological and interpretational aspects require further clarification and validation to strengthen the scientific rigor and biological relevance of the study's conclusions.

Q#1) What is the relation between colony health and the gut microbiota of honeybees?

Thank you for this comment. We answered this question in lines 65 to 69:” Indeed, by improving bee health at the individual level, a balanced microbiota also improves colony performance, since healthy bees are more numerous (less mortality), have fewer hygienic tasks to perform (e.g. evacuating dead individuals) and can therefore allocate their full capacity to carrying out main tasks of the colony, including pollination and honey production.”.

Q#2) In the abstract, authors are required to show the effect of the combination of Bactocell and Fumidil on the colony health.

Thank you for your suggestion, these effects have been added line 21: “The combination of these products restored the microbiota close to control, but with mixed results for colony performance.”.

Q#3) In the introduction line 34, the authors are advised to give more examples of these biotic and abiotic stressors.

Thank you for this comment, more examples of stressors have been added line 34-35: “These stressors may be biotic (e.g. Varroa destructor parasite and deformed wing virus) or abiotic (e.g. antifungals, pesticides)”.

Q#4) Can you give more details about the type of probiotic bacteria present in Bactocell and whether it is only 1 type of bacteria or more than one type?

Thank you for this comment, we added information about the probiotic in the introduction line 92: “This probiotic consists of the bacterial strain Pediococcus acidilacticis. This bacterium is a member of lactic acid bacteria (LAB), able to adapt to the intestinal environments of several types of host.”.

Q#5) In line 45, what is meant by firm 4 and firm 5?

This part has been specified lines 47-48: “Bombilactobacillus sp. (formerly named Lactobacillus Firm-4), Lactobacillus sp. (formerly named Lactobacillus Firm-5)”.

Q#6) In lines 69 and 70, based on references 14 to 16, what is the new added value this research will highlight?

Thank you for your guidance, we added a few lines in this sense lines 78-80: ”Therefore, this study could also highlight the effects of gut dysbiosis on the performance of functions provided by the microbiota.”.

Q#7) In lines 97 to 102, this part is a conclusion and should not be here in the introduction part.

This part aims to present our three hypotheses. To address your concern, we added a sentence line 108 (“Our working hypothesis was that both treatments and their combination would change microbiota structure and correlate with differences regarding hive performance after wintering. Specifically, three hypotheses were tested: 1) Fumagilin will alter the microbiota composition and its resiliency (dysbiosis) and decrease colony performance; 2) Bactocell® will maintain the microbiota composition and its resiliency (eubiosis) and enhance colony performance and 3) combining Fumidil-B® and Bactocell® (group FB) will maintain or restore the microbiota composition and its resiliency (eubiosis) without having a negative impact on colony performance.”).

Q#8) The ethical approval number is missed.

While the authors understand the concern of the reviewer, “No animal health care permits were required for this research.” as mentioned line 115.

Q#9) In line 107, please include the year 2021 after July 5th. Also, this date is different from that mentioned in Figure 1. It is the 15th of July 2021 in Figure 1. Please double-check the correct one.

Thank you for this comment, the year have been added and the proper date corrected line 118.

Q#10) In lines 141 – 143, the numbers containing 10 to the power need to be adjusted.

Thank you, all occurrences have been corrected in the Table 1 and in the text.

Q#11) In line 227, please mention the manufacturer country after the company name.

The manufacturer have been added line 245: “(Beckman Coulter Genomics, Mississauga, ON, Canada)”.

Q#12)  In line 256, the authors mentioned that they used SILVA Database, but which version did they use, and did the authors make any training for classifiers?

As suggested, we specified the version of silva line 277 (“SILVA 138.1 database (last update, July 2020)”).

For the annotation process, we employed the DADA2 package functions assignTaxonomy and addSpecies. The file silva_nr99_v138.1_wSpecies_train_set.fa used with the assignTaxonomy function is a pre-trained classifier provided by the DADA2 developers. This classifier was trained using the SILVA reference sequences from version 138.1. Therefore, we did not perform custom training for the classifier; instead, we relied on the pre-trained classifier for taxonomic assignment. This was specified in lines 277 to 280: “The annotation was performed using the functions “assignTaxonomy” and “addSpecies” (package DADA2). A pre-trained classifier provided by the DADA2 developers (the file silva_nr99_v138.1_wSpecies_train_set.fa) was used with the functions “assignTaxonomy”.”.

Q#13) P value, the P letter should be italicized all over the manuscript.

This has been corrected throughout the manuscript.

Q#14) Many other data could be grasped from the sequencing microbiome data which should be useful like ANCOM or LEFSE. Moreover, the authors didn't perform PICRUST to predict metabolic pathways differences between groups.

We carried out a LefSE analysis which did not show any significant results. We chose not to mention it in the manuscript because of the density of the part of the results you mentioned.

We did not perform a PICRUST analysis, as functional inference can be highly speculative for non-human, or at least non-mammalian, organisms. Indeed, the reference database of complete bacterial genomes is heavily biased in favor of the human gut microbiota (see for instance Languille, 2018 and Matchado et al. 2024).

Moreover, any given bacterial “species” inferred from a short 16S rRNA gene sequence can actually encompass different strains with different functional capabilities, due to horizontal gene transfer for instance, and therefore occupy different ecological niches.

In addition, annotations of bacterial genes are still notoriously inaccurate, making biological interpretations of microbiome community function uncertain. Moreover, we do not know whether these genes are transcribed or translated. This further limits the relevance of inferred functions. Conclusions on microbiome function derived from PICRUSt (or any other similar tool) must be considered highly speculative and undoubtedly require further validation through functional assays.

Q#15) In line 416, the wrong spelling of Pediococcus sp. should be corrected.

Thank you, it has been corrected throughout the manuscript.

Q#16) What about microbiome data deposition and availability?

This information is specified in the Data Availability Statement (line 770) : “The data are available in the NCBI Genbank® with the accession number PRJNA1103710 (https://www.ncbi.nlm.nih.gov/nuccore/?term=PRJNA1103710).”

Q#17) The results section is dense and lacks clarity in presenting key findings. The authors should consider reorganizing the results to improve readability.

Thank you for this comment. We improved the clarity by reducing the amount of text throughout the result section and by restructuring and precising the different parts of this section.

Q#18) The interpretation of results is often ambiguous, with the authors failing to clearly link findings to underlying biological mechanisms or theoretical frameworks. 

Our results were interpreted cautiously. No speculative discussions were undertaken on the putative functions of single bacterial taxa, with the exception of some honey bee core microbiota taxa, for which functions are well documented and supported in the literature.

Q#19) The manuscript contains numerous typographical errors and grammatical mistakes that detract from its overall professionalism. A thorough proofreading and editing process is needed to correct these issues.

We apologize for this issue. The MS was thoroughly proofread. 

Q#20) How do the authors ensure the robustness of their experimental design, particularly regarding sample size and randomization procedures?

We took several steps to maximize the robustness of our study. First, we selected a high number of colonies per group, with 12 colonies per group, which is higher than the sample sizes used in similar studies conducted in apiaries (see Bleau et al., 2023, and Ribière et al., 2019 for example). This larger sample size enhances the statistical power of our study and increases the reliability of our findings. Additionally, we randomly allocated colonies to each group to minimize selection bias (line 146). The only constraint in this randomization process was to ensure that the groups were of equal strength, helping to control for initial differences between the colonies. Moreover, to control for potential genetic impacts, the colonies came from sister queens (line 117). Finally, the colonies were placed randomly in two apiaries to minimize the impact of environmental factors (line 149). This approach ensures that any observed effects can be attributed to the treatments rather than pre-existing differences between the groups. By carefully determining our sample size and implementing randomization procedures, we have taken steps to ensure the robustness and reliability of our experimental design. 

Q#21) To what extent do the findings of this study align with or diverge from prior research on similar topics? Discuss whether the observed shifts in microbial diversity are consistent with expectations based on known ecological principles and previous research findings in honey bee microbiome studies.

Thank you for this comment. A study also carried out in apiaries and with the same probiotic provided similar results (increase in the number of capped cells) (see Bleau et al., 2023). Regarding the microbial aspects, the signs of dysbiosis linked to the exposure to the Fumidil B® have also been observed with other antibiotics as mentioned line 689-691: (“Overall, significant impairments of the honey bee microbiota have been repeatedly associated with the use of antibiotics such as tetracycline-derived compounds [7,51].”). We added reference line 696 to support the results obtained by the combination of products on the microbiota (“A recent study has also showed that Bactocell® restored the bee microbiota to a state close to control after induction of dysbiosis by a stressor factor (V. ceranae) [54].”) and line 722 (“Furthermore, alpha diversity data showed that Bactocell® group exhibited reduced evenness, accompanied by a decrease of rare taxa (e.g. Arsenophonus, Enterobacter), while some core taxa abundance remained stable or increased, suggesting an antagonistic activity of the probiotic against bee opportunistic pathogens (e.g. Paenibacillus genus), as observed in the literature [55].”).

Q#22) How do the authors interpret the observed changes in alpha and beta diversity indices within and between groups?

Thank you for this question. There was no change regarding beta diversity and some hypotheses are proposed is the discussion regarding this topic (subsection 4.4 of the discussion). As developed in the subsection 4.3 of the discussion, the decrease of evenness observed for all groups except the Fumidil B® reflects the development of a structured microbiota “where the rare taxa decreased (Arsenophonus, Enterobacter for example) while the core taxa increased or remained unchanged (Commensalibacter, Lactobacillus, Bombilactobacillus).” (lines 613-615) while “the increasing of evenness indicates a more even distribution of species abundances, that might reflect a dysbiosis in the microbiota following the Fumidil B® exposure since core species decreased in abundance (Commensalibacter) while rare taxa increased […]” (lines 615-618).

Q#23) What are the potential implications of the study's findings for honey bee health management?

Thank you for your interesting question. We addressed it in a new subsection at the end of the discussion (see lines 740 to 745:” This study represents a first step towards helping honey bees adapt to exposure to chemical molecules and hindering the multiplication of hive pathogens, with a probiotic solution that is applicable and available to beekeepers. It also highlights the key role played by the intestinal microbiota in bee survival and colony performance. Finally, it paves the way for research into a global solution to improve bee health in anticipation of exposure to multiple stressors.”).

Q#24) Can the authors clarify the biological mechanisms underlying the observed interactions within the microbiota networks?

Thank you for your valuable comment, we clarified this topic lines 649-653: “When an increase in abundance of one strain is associated with an increase in abundance of another, it indicates a positive interaction. Conversely, if such increase corresponds with a decrease in abundance of the other, the interaction is negative. If there is no correlation between the abundance variations of the two taxa, the interaction is considered neutral and is not represented in the network.”. The biological meaning of these interactions and overall network topology are specified from line 657: “High connectivity combined with high rate of positive interactions confers greater resilience to microbiota, since a key function could be performed by several taxa (i.e. redundancy) and thus secured in case of disappearance of one taxon [47,48]. On the opposite, an increased number of negative interactions is interpreted as a hallmark of dysbiosis [12,49,50].”.

Q#25) Are there any additional analyses or experiments that the authors believe would strengthen the study's conclusions?

Thank you for this question. We added a section at the end of the discussion to answer this, line 734: “In the future, it may be appropriate to optimize the doses or methods of probiotic supplementation (mixing with pollen patty or providing continuous supplementation during wintering, for example). In addition, it would have been interesting to carry out transcriptomic analysis to assess the functional impact of the tested products. Similar tests carried out in a controlled environment (i.e. cage-bees) could also be relevant for understanding the underlying mechanisms associated with the products tested.”.

Q#26) Address the potential functional roles of the microbial communities identified, considering their taxonomic composition and known metabolic capabilities. Discuss any functional predictions or hypotheses derived from the taxonomic data.

As stated above, we did not perform PICRUST analysis as functional inference can be highly speculative for the reason detailed in our previous answer.

Q#27) Provide detailed information regarding the exact sampling protocol followed, including the specific anatomical sites sampled from honey bees, the sampling frequency, and any precautions taken to minimize environmental contamination during sampling.

The requested precision on sampling have been added in lines 183: “Approximately 100 honey bees were sampled in each hive, once before the treatments (t0, October 7) and once after the fall treatments (t21, October 28). After securing the queen, bees were collected from a broodless frame in a 50 ml Eppendorf tube. Samples were immediately put on dry ice and stored at -86°C until lab analysis (Thermofisher -86°C FORMA 908, Waltham, MA, USA). Sampling was carried out twice, before exposure to the treatments (t0) and after 21 days (t21).”.

The anatomical site of interest and the precautions taken to minimize contamination were specified in lines 231: ”The sampled bees were dissected under flame and with alcohol-sterilized instruments to prevent external contamination. After collecting the midgut, DNA was extracted using the salt method described by Aljabani and Martinez (1997).”.

Q#28) Discuss how changes in microbial composition may impact honey bee health and colony productivity over time.

Thank you for your suggestion. We addressed this topic lines 730 to 738: “To conclude on the biological significance of the main results generated by the present study, we would like to recall that the changes detected in the composition of the microbiota can have positive or negative impacts on bee health at the individual level. These compositional changes can either prevent (e.g. increased colonization resistance) or favor (e.g. decreased colonization resistance) the establishment of opportunistic and potentially pathogenic taxa. Moreover, they can promote or alter certain functions essential to bees that depend on particular taxa. Thus, by affecting bee health, variations in microbiota composition also influence the productivity of their colonies, as healthier bees are more capable of fully performing the colony main tasks.”.

Round 2

Reviewer 3 Report

Comments and Suggestions for Authors

Recommend to accept.